# Drip Fertigation Increases Maize Grain Yield by Affecting Phenology, Grain Filling Process, Biomass Accumulation and Translocation: A 4-Year Field Trial

**DOI:** 10.3390/plants13141903

**Published:** 2024-07-10

**Authors:** Ruiqi Du, Zhijun Li, Youzhen Xiang, Tao Sun, Xiaochi Liu, Hongzhao Shi, Wangyang Li, Xiangyang Huang, Zijun Tang, Junsheng Lu, Junying Chen, Fucang Zhang

**Affiliations:** 1Key Laboratory of Agricultural Soil and Water Engineering in Arid and Semiarid Areas of Ministry of Education, Northwest A&F University, Yangling 712100, China; duruiqi@126.com (R.D.); youzhenxiang@nwsuaf.edu.cn (Y.X.); 2021050986@nwsuaf.edu.cn (T.S.); 2023055903@nwafu.edu.cn (X.L.); shihongzhao7@nwafu.edu.cn (H.S.); 2022050965@nwsuaf.edu.cn (W.L.); 2023055900@nwsuaf.edu.cn (X.H.); tangzijun@nwsuaf.edu.cn (Z.T.); junyingchen@nwsuaf.edu.cn (J.C.); 2Institute of Water–Saving Agriculture in Arid Areas of China, Northwest A&F University, Yangling 712100, China; 3State Key Laboratory of Herbage Improvement and Grassland Agro-Ecosystems, College of Ecology, Lanzhou University, Lanzhou 730000, China; junshengup@163.com

**Keywords:** grain filling, biomass accumulation, yield components, photosynthetic rate, drip fertigation

## Abstract

Drip fertigation (DF) is a widely used technology to increase grain yield with water and fertilizer conservation. However, the mechanism of high grain yield (GY) under DF is still unclear. Here, a four-year field experiment assessed the impacts of four treatments (i.e., conventional irrigation and nitrogen application, CK; drip irrigation with conventional nitrogen fertilization, DI; split-nitrogen fertigation with conventional irrigation, SF; and drip fertigation, DF) on maize phenology, leaf photosynthetic rates, grain filling processes, plant biomass, and GY. The results showed that DF significantly increased maize GY by affecting phenology, grain filling traits, aboveground biomass (BIO) accumulation, and translocation. Specifically, DF significantly increased leaf chlorophyll content, which enhanced leaf photosynthetic rates, and together with an increase of leaf area index, promoted BIO accumulation. As a result, the BIO at the silking stage of DF increased by 29.5%, transported biomass increased by 109.2% (1.2 t ha^−1^), and the accumulation of BIO after silking increased by 23.1% (1.7 t ha^−1^) compared with CK. Meanwhile, DF prolonged grain filling days, significantly increased the grain weight of 100 kernels, and promoted GY increase. Compared with CK, the four-year averaged GY and BIO increased by 34.3% and 26.8% under DF; a 29.7%, 46.1%, and 24.2% GY increase and a 30.7%, 39.5%, and 29.9% BIO increase were contributed by irrigation, nitrogen, and coupling effects of irrigation and nitrogen, respectively. These results reveal the high yield mechanism of drip-fertigated maize, and are of important significance for promoting the application of drip fertigation.

## 1. Introduction

The global food crisis caused by population growth and climate change has attracted widespread attention [1,2]. It is reported that the output from the agricultural system must be doubled by 2050 [3], posing a significant challenge to agricultural production. Maize (*Zea mays* L.) is one of the most widely cultivated food and feed crops in the world, which is of great significance in ensuring food security. As the world’s second-largest maize producing country, China produces 22.4% of the world’s maize yield with a 20% planting area [4]. However, the grain yield per unit area in China (6.3 t ha^−1^) still has a certain gap compared to developed countries such as the United States (10.5 t ha^−1^, [5]). This indicates that the grain yield of maize in China can be highly improved. Narrowing this gap is of great significance for addressing global food security.

Crop breeding may help to increase maize production to a certain extent, but from the perspective of farmers, water and fertilizer management is the most direct and effective practice [6], especially in arid and semi-arid regions where water and nitrogen restrict high maize yields. In order to achieve a high grain yield, farmers often apply excessive amounts of water and nitrogen [7]. This may lead to leaching loss of nitrate nitrogen and an increase in greenhouse gas emissions [8,9], further exacerbating global warming. Conversely, the intensification of global warming has increased the frequency of extreme rainfall events [10], making it difficult for traditional water and fertilizer management practices to achieve high crop yields and efficiently utilize water and fertilizer.

Drip fertigation is known as the most advanced water and fertilizer management technology due to its exciting and flexible water and fertilizer management concept; it significantly improves crop yield and water and fertilizer utilization efficiency and is widely used in field crop production [11,12]. Many studies have explored the response of maize grain yield to drip fertigation and concluded that drip fertigation significantly improved maize grain yields with higher water and fertilizer utilization efficiency [9,13]. A meta-analysis in China showed that drip fertigation significantly increased maize grain yields 12.7% compared to famers’ practices [14]. And another global meta-analysis also reported that drip fertigation increased the average yield by 20% and increased nitrogen use efficiency and water productivity by 26% and 51%, respectively [15]. The above studies have all demonstrated that drip fertigation can significantly improve maize grain yields and water-nitrogen use efficiency, but they have not revealed the high-yield mechanism of drip fertigation for maize, nor have they clarified the contribution of irrigation, nitrogen, and their coupling effects under drip fertigation conditions to maize yield increases.

Previous studies have reported that crop yield is determined by pre-flowering dry matter transport and post-flowering dry matter accumulation [16,17]. Transporting more pre-flowering dry matter to grains and increasing the accumulation of post-flowering dry matter are effective ways for higher crop yields [18,19]. The final grain yield of maize was determined by grain weight, grain number per ear, and ear number per area. At a suitable planting density, grain number per ear and grain weight are key factors determining maize grain yields [20], which are directly influenced by water and nitrogen management [21,22]. The newly formed carbohydrates after flowering account for 60–90% of the dry matter in grain yields [23], indicating that the grain filling process is crucial for high crop yields. Some researchers have reported that the grain filling rate and duration affect the expansion of grain storage capacity [24,25], which is regulated by water and fertilizer management [26,27]. However, there is currently no research proving how grain filling parameters respond to drip fertigation, and whether the increased maize yield under drip fertigation is achieved by the changed grain filling process and optimized dry matter accumulation and distribution.

Based on the above analysis, we hypothesize that drip fertigation enhances maize grain yield mainly by (1) changing the grain filling process (prolonging the grain filling duration and increasing the grain filling rate) and (2) optimizing dry matter accumulation and distribution. Therefore, we conducted a four-year field experiment to (1) explore the response of maize phenological phases, leaf chlorophyll content, and photosynthetic parameters to drip fertigation; (2) investigate the grain filling process of summer maize and the characteristics of above-ground biomass accumulation and transportation; and (3) quantify the contribution of irrigation, nitrogen fertilization, and the coupling effects of irrigation and nitrogen to yield increases. This study can provide a better understanding for the high-yield mechanism of summer maize under drip fertigation conditions.

## 2. Materials and Methods

### 2.1. Experiment Site Description

The field experiment was conducted during the 2017–2020 summer maize growing seasons at the Yangling Agriculture High-Tech Industrial Demonstration Zone (34°20′ N, 108°24′ E, 506 m), Shannxi province, China. This region is located in a temperate continental monsoon climate area, and the annual mean temperature was 13.3 °C. The annual evaporation and precipitation were 1500 mm and 560 mm, respectively [12]. The soil texture was characterized by silty clay loam based on the USDA soil classification system. The basic properties of the plough layer soil are presented in Appendix A.

### 2.2. Experiment Design

The field experiment consisted of four treatments: (1) drip fertigation, DF; (2) split fertilization with conventional (flood) irrigation, SF; (3) drip irrigation with conventional (one-time fertilization) fertilization, DI; and (4) conventional irrigation and fertilization, CK. In this study, a drip irrigation system was used for drip fertigation, drip irrigation, and split fertilization. Each treatment was replicated three times, for a total of 12 plots (25 m^2^, 7.0 × 3.6 m). The nitrogen (210 kg ha^−1^) was fertigated for DF and SF according to the fertilizer requirement of summer maize, i.e., 20%, 30%, 30%, and 20% nitrogen was fertilized at sowing, seedling, jointing and grain filling stages, respectively. The total nitrogen (210 kg ha^−1^) was applied once with precipitation for DI and CK after maize sowing. All treatments received identical P_2_O_5_ (90 kg ha^−1^) and K_2_O (60 kg ha^−1^) with precipitation after maize sowing. For SF and CK, only 90 mm water was irrigated (flood irrigation) at the jointing stage, while a drip irrigation system was used for DF and DI irrigation according to the precipitation distribution. Specifically, 30 mm water was irrigated when the accumulated ET_c_ (crop evapotranspiration, calculated according to [28,29]) reached or was close to 30 mm and no precipitation occurred in the next 3 days (based on the weather forecast). The irrigation amounts for DF and DI were 180, 120, 90, and 60 mm in the 2017, 2018, 2019, and 2020 summer maize growth periods, respectively. The detailed irrigation and fertilization times can be found in Appendix A. Summer maize cv. ‘*Zhengdan 958*’ was planted at 70,922 plants·ha^−1^ (plant spacing 23.5 cm and row spacing 60 cm). In this study, only one ear was produced in each plant. Other field management practices, including spraying and weeding, remained consistent with local practices.

### 2.3. Data Collection

#### 2.3.1. Meteorological Data and Crop Phenology

Daily maximum and minimum temperatures, precipitation, sunshine hours, humidity, and wind speed during the experiment periods were recorded using an automatic weather station to calculate reference crop evapotranspiration according to [28]. Date of emergence, 6-leaf, tasseling, silking, blister, and maturity in each plot were accurately observed and recorded during each growing season. When 50% of the plants in each plot exhibited characteristics of a certain growth period, they were determined to be entering that growth period [30]. These data were used to calculate the duration of specific growth stages.

#### 2.3.2. Leaf Area Index, Chlorophyll Content, and Photosynthetic Characteristics

At the tasseling (VT) and dough (R4) stages, three plants were selected to measure the green leaf area of summer maize using a ruler (the accuracy is 0.1 cm), and the leaf area index (LAI) was calculated by the formula LAI = (leaf length (m) × maximum leaf width (m) × 0.75 × planting density)/10,000. The ear leaf at VT and R4 stages was used to measure chlorophyll content (Chl__VT_ and Chl__R4_) and the photosynthetic rate (Pn__VT_ and Pn__R4_). About 0.1 g of fresh leaf was weighed and chlorophyll extracted using 95% ethanol. A Genesys 10 UV spectrophotometer was used to determine chlorophyll a and b content at 665 and 649 nm [31]. The sum of chlorophyll a and b was used to represent Chl. At a.m. 9:00–11:00, the photosynthetic characteristics were measured using a Li-6400 handheld photosynthesis system under a clear sky at the VT and R4 stages.

#### 2.3.3. Aboveground Biomass, Grain Yield, and Yield Components

At the silking and maturity stages, three plants in each plot were selected to measure aboveground biomass (BIO). The summer maize at the silking stage was separated as stem and leaf, and was separated as stem, leaf, bract, grain, and rachis at the maturity stage. The weight of these components was record after being oven-dried to a constant weight. The BIO accumulation after silking (BIO_as_) was calculated by the formula BIO_as_ = BIO at maturity (BIO_m_) − BIO at silking (BIO_s_). The transfer amount of BIO (TA_bio_) was calculated by the formula TA_bio_ = BIO_s_ − (BIO_m_ − GY). The transfer rate of BIO (TR_bio_) was calculated by the formula TR_bio_ = TA_bio_/BIO_s_. The contribution of TA_bio_ to grain yield (C_t_) was calculated by the formula C_t_ = TR_bio_/GY. The contribution of BIO_as_ to grain yield (C_a_) was calculated by the formula C_a_ = BIO_as_/GY. At summer maize maturity, fifteen maize ears were randomly collected in the middle of each plot to determine grain yield and weighed (up to 14% moisture content) [12]. Ear length, ear diameter, bald tip length, row numbers per ear, kernel numbers per row, and grain weight of 100 kernels were the statistics for each ear.

#### 2.3.4. Grain Filling Process

At the beginning of the silking stage, 40 uniform plants were tagged in each plot, and 3 maize ears were selected every 5–7 days to peel off 30, 40, or 30 niblets at 1/3, 1/2, and 2/3 of each ear, respectively. The niblets were dried at 80 °C to a constant weight and weighed using a millionth scale. The grain filling process was assessed using logistic Equation (1). The first derivative of the logistic equation was used to calculate the grain filling rate as Equation (2). The maximum growth weight (W_max_) was obtained using Equation (3). The maximum grain filling rate occurrence time (t_m_) was obtained using Equation (4). The maximum grain filling rate (G_max_) was calculated using Equation (5).
(1)$$y=\frac{a}{1+\mathrm{bexp}(-\mathrm{cx})}$$
(2)$$y^{\prime }=\frac{{\mathrm{abce}}^{-\mathrm{cx}}}{{{(1+\mathrm{be}}^{-\mathrm{cx}})}^{2}}$$
(3)$$W_{\max}=\frac{a}{2}$$
(4)$$t_{m}=\frac{\ln(b)}{c}$$
(5)$$G_{\max}=(c\times W_{\max})/(1-\frac{W_{\max}}{a})$$
where a represents the final 100-kernel dry grain weight, g, b and c are estimated parameters, and $y^{\prime }$ represents the grain filling rate.

### 2.4. Statistical Analysis

Aboveground biomass accumulation, leaf area index, chlorophyll content, photosynthetic characteristics, grain yield, 100-kernels weight, rows per ear, kernels per row and grain filling traits were compared with a least significant difference (LSD) test at *p* = 0.05 level. The statistical analysis was carried out using R 4.2.2 software.

## 3. Results

### 3.1. Maize Phenological Phase

Daily precipitation, average temperature and reference crop evapotranspiration during summer maize growing seasons (2017–2020) are presented in Appendix A. The average temperature, total precipitation, and reference crop evaporation in the summer maize growing seasons were 25.0 °C, 235 mm, and 395.0 mm in 2017; 24.2 °C, 417 mm, and 351.0 mm in 2018; 23.0 °C, 507 mm, and 331.9 mm in 2019; and 22.8 °C, 526 mm, and 392.9 mm in 2020, respectively. A drought period occurred at the vegetative stage of 2017, V6-VT stage of 2018, R2-R4 stage of 2019, and R4-R5 stage of 2020, respectively.

Different water and nitrogen management significantly affected the phenology of summer maize (Figure 1). Split nitrogen fertilization (DF and SF) increased the growth period (days) under four growth seasons, and the 4-year averaged growth days of DF and SF were significantly higher than those of CK and DI (Figure 1a). Both DF and SF had the same growth days (103.5 d), which is 2.7 days greater than CK. The grain filling period, especially, was extended by 3.1 and 4.3 days, respectively. DF and SF significantly decreased the proportion of the vegetative growth period (EM-V19) but significantly increased the proportion of grain filling period (Figure 1b). Compared with CK, the 4-year averaged proportion of the grain filling period significantly increased by 1.9% and 3.1% under SF and DI, respectively.

### 3.2. Leaf Area Index, Chlorophyll Content, and Photosynthetic Performance

Irrigation, nitrogen, and growing year had significant effects on the leaf area index (LAI), chlorophyll (Chl) content, and photosynthetic (P_n_) rate (*p* < 0.01, Table 1). The interaction of irrigation and nitrogen and irrigation and year showed a highly significant impact on the summer maize LAI (*p* < 0.01) but had an insignificant impact on Chl content and the P_n_ rate (*p* > 0.05). The interaction of nitrogen and growing year had a highly significant effect on the LAI at the tassling stage (*p* < 0.01), and the interaction of irrigation, nitrogen, and growing year had a significant effect on Chl content only in the R4 stage (*p* < 0.05).

Drip irrigation and split fertigation significantly increased the LAI, Chl content, and Pn rate of summer maize (Figure 2). The highest LAI, Chl, and Pn were observed under DF. And the 4-year averaged value of LAI, Chl, and Pn increased by 42.7%, 10.6%, and 24.9% at the tasseling stage, and increased by 86.9%, 23.8%, and 42.0% at the grain filling stage, respectively, under DF compared with the CK treatment. In the 2017, 2018, and 2019 growing seasons, DI and SF showed an insignificant difference in LAI, while the LAI of SF was significantly higher than DI in the 2020 growing season (Figure 2a,b). Both SF and DF significantly increased the Chl and Pn of summer maize compared with CK and DI (Figure 2c–f). Additionally, the Chl and Pn of DF were significantly higher than that of SF, and DI was higher than that of CK.

### 3.3. Grain Filling Characteristics

DF significantly increased the grain filling rate of summer maize, especially in the middle and late stages of the grain filling processes (Figure 3, Table 2). The grain weight of the maximum grain filling rate (W_max_), the occurrence time of the maximum grain filling rate (T_max_), the maximum grain filling rate (V_max_), and the average grain filling rate (V_ave_) of summer maize showed different trends. The four-year averaged W_max_ of SF and DF were significantly higher than those of CK and DI. The four-year averaged T_max_ of DF was higher than other treatments, but the difference did not reach a significant level (*p* > 0.05). The four-year averaged V_max_ of DF was significantly higher than other treatments, while the V_ave_ of DF had an insignificant difference with other treatments (Table 2).

DF and SF significantly increased the W_max_ of summer maize in the 2017 and 2020 growing seasons (Table 2). DF significantly increased W_max_ in 2019, while W_max_ had an insignificant difference among treatments in the 2018 growing season. In the 2017 growing season, DF and SF significantly increased the T_max_ of summer maize compared with CK, but T_max_ did not show a significant difference in the 2018, 2019, and 2020 growing seasons. DF significantly increased the V_max_ of summer maize, while the V_ave_ of DF had an insignificant difference with other treatments in the 2017–2020 growing seasons.

### 3.4. Grain Yield and Yield Components

Irrigation, nitrogen, and growing year significantly (*p* < 0.05) affected grain yield and yield components (Table 3). The interaction of irrigation and nitrogen had highly significant effects on rows per ear, kernels per row, bare top length, and grain yield, and had significant effects on 100-kernel grain weight and ear length. The interaction of nitrogen and year had highly significant (*p* < 0.01) effects on kernels per row and bare top length. The interaction of irrigation and year had highly significant effects on ear rows, grains per row, ear length, and bare top length, and had significant effects on grain yield. The interaction of irrigation, nitrogen, and year had highly significantly effects on ear diameter and bare top length.

Overall, DF significantly increased grain yield and yield components (except for bare top length) compared with DI, SF, and CK (Table 4). Except for the fact that DI had a higher ear length than SF, there was an insignificant difference in summer maize grain yield and yield components between DI and SF; both were significantly higher than those of CK (except for bare top length). Compared with CK, grain yield (4-year averaged) of DF was increased by 10.2%, 15.8%, and 34.3% (with 29.7%, 46.1%, and 24.2% contributed by irrigation, split nitrogen, and the coupling effect of irrigation and nitrogen) (Table 5). Rows per ear increased by 2.1%, 0.2%, and 7.2%; kernels per row increased by 5.2%, 7.6%, and 19.3%; and 100-kernel grain weight increased by 0.9%, 2.8%, and 7.3% of DI, SF, and DF, respectively, compared with CK (Table 4). However, significant differences in grain yield and yield components of summer maize under different growing years were noticed. Specifically, DI and DF significantly increased rows per ear, kernels per row, and ear length compared with SF and CK in 2017 (Table 4). Ear diameter and grain yield of DF were significantly higher than SF, DI, and CK, while SF and DI had insignificant effects in those indicators. In 2018, DF and DI significantly increased rows per ear compared with CK and SF, and DF was significantly higher than SF, DI, and CK in other indicators (except for bare top length). In 2019, DF significantly increased rows per ear, grains per row, 100-kernel grain weight, and ear length compared with the other three treatments. Grain yield followed a trend where DF > SF > DI > CK, but DF and SF had insignificant differences. In 2020, the grain yield of DF and SF had insignificant differences, but both were significantly higher than CK.

### 3.5. Aboveground Biomass Accumulation and Translocation

Irrigation, nitrogen and growing year had a significant effect on aboveground biomass at maize maturity (BIO_m_), aboveground biomass at maize silking (BIO_s_), accumulation of aboveground biomass after the silking (BIO_as_), and transport amount and rate of BIO (TA_bio_ and TR_bio_) (Table 3). The interaction of irrigation and nitrogen and irrigation and growing year also had a significant effect on BIO_m_, BIO_s_, and TAbio. The interaction of irrigation, nitrogen, and growing year had a significant effect on TAbio and TRbio.

DF also significantly increased BIO_as_; the 4-year averaged BIO_as_ of DF significantly increased by 9.8%, 14.4%, and 23.1% compared with SF, DI, and CK, respectively (Table 6). Correspondingly, DF, SF, and DI also significantly increased TA_bio_ and TR_bio_; the 4-year averaged TA_bio_ of DF, SF, and DI was 2.2, 1.5, and 1.4 t ha^−1^, which increased by 109.2%, 40.6%, and 27.2%, respectively, compared with CK (1.1 t ha^−1^). The 4-year averaged TR_bio_ of DF, SF, and DI was 17.9%, 13.9%, and 12.9%, which was significantly higher than that of CK (11.0%). DF significantly increased the contribution of TA_bio_ to grain yield (C_t_), while the C_t_ showed an insignificant difference in SF, DI, and CK. The C_t_ of DF increased by 7.4%, 5.3%, and 4.7% compared with CK, DI, and SF, respectively.

The accumulation and translocation of BIO and their contribution to grain yield under different irrigation and nitrogen amounts vary among different years (Table 6). Specifically, BIO_m_, BIO_s_, BIO_as_, TA_bio_, TR_bio_, and C_t_ followed DF > DI > SF > CK in 2017, followed DF > DI ≌ SF > CK in 2018, followed DF > SF > DI ≌ CK in 2019, and followed DF ≌ SF > DI > CK in the 2020 growing year. It is clear that, except for C_a_, DF had significantly higher values than CK and DI in the four years.

### 3.6. Correlation Analysis

Correlation analysis showed that BIOs had a highly significant positive correlation with LAI__VT_ and Pn__VT_, but had a significant negative correlation with SD1 (Figure 4). BIO_as_, BIO_m_, and GY had a highly significant positive correlation with RNE, GW__100_, KR, LAI__R4_, LAI__VT_, TA_bio_, TR_bio_, V_max_, and SD4, but had a highly significant positive correlation with BTL and SD1. RNE, GW__100_, and KR had a highly significant positive correlation with LAI__R4_, LAI__VT_, Pn__VT_, and Pn__R4_.

## 4. Discussion

### 4.1. Response of Maize Phenology to Drip Fertigation

In this study, DF significantly prolonged the proportion of the grain filling period (R2–R6) and shortened the proportion of the vegetative period (EM-V19) of summer maize, especially in 2017 (dry year) (Figure 1b). Meanwhile, whole growth days also increased under DF and SF compared with CK (Figure 1a). This difference can be explained from the following two aspects. One is that the soil moisture remains in a relatively sufficient state under DF treatment [32], which avoids maize suffering drought stress. Drought stress significantly increased the vegetative period and decreased the grain filling duration, and this impact increased with the increase of stress levels [33]. Another reason was that the split-nitrogen application increased the chlorophyll content of the grain filling period (Figure 2d), avoiding nitrogen stress on the maize. Drought stage and physiological maturity can be advanced when maize suffers nitrogen stress [34]. In this study, nitrogen was applied based on the nutrient demand of summer maize under DF and SF treatments, ensuring a sufficient nitrogen supply during the grain filling period and effectively avoiding the early-senescence of maize leaf (higher LAI at R4 stage, Figure 2b). Therefore, the grain filling period with DF and SF was significantly prolonged compared to with CK.

### 4.2. Accumulation and Translocation of Aboveground Biomass

The accumulation of aboveground biomass is a prerequisite of high crop yields. Promoting the transport of pre-flowering biomass to grains and increasing post-flowing biomass accumulation are effective ways for high crop yields [35,36]. In this study, DI, SF, and DF significantly increased pre-flowing biomass by 9.0%, 11.0%, and 29.5%, and increased post-flowing biomass by 7.7%, 10.0%, and 23.1%, and the biomass transport amount for CK, DI, SF, and DF was 1.1, 1.4, 1.5, and 2.2 t ha^−1^, respectively (Table 6). A possible reason for this is that drip irrigation significantly reduced water loss by reducing seepage and evaporation [37,38] and provided a suitable moisture environment for crop growth. Drip irrigation promoted the accumulation of biomass under the same irrigation amounts, and drip irrigation with less water does not significantly reduced biomass accumulation [39,40]. Optimizing nitrogen fertilizer applications can significantly increase aboveground biomass accumulation to sustain high grain yields under nitrogen fertilizer reduction [9,29]. In this study, DF not only significantly increased BIOs and BIOas compared to CK, but also significantly increased BIOs and BIOas compared with DF and SF, and the increased BIO of DF was significantly higher than that the sum of increased BIO of DF and SF. This indicates a positive coupling/promoting effect exists between water and nitrogen. Our results show that irrigation, nitrogen, and the coupling effect of irrigation and nitrogen contributed a 30.7%, 39.5%, and 29.9% BIO increase under DF compared with CK (Table 5). This clearly demonstrates the contribution of water–fertilizer coupling to increased BIO.

### 4.3. Response of Grain Filling Traits and Yield Components to Drip Fertigation

The rate and duration of grain filling are the main factors affecting grain weight [41,42]. In this study, the grain filling rate of DF was higher than the other treatments, especially in the middle and later periods of the grain filling process (Figure 3). The maximum grain filling rate of DF was significantly higher than other treatments during the 2017–2020 growing seasons, while the average grain filling rate showed insignificant differences among the four treatments except for SF in 2019 (Table 2). These results demonstrate that DF changed the grain filling process but had an insignificant effect on the average grain filling rate. Therefore, the increased 100-kernel grain weight was mainly attributed to the prolonged duration of grain filling (Figure 1).

In this study, grain yield components (rows per ear, kernels per row, and 100-kernel grain weight) presented different trends in different years (Table 5). The main reason was due to changes in the precipitation distribution across the different years. Previous research has reported that drought stress occurring at different growth stages will affect different yield components. The drought stress at the jointing and anthesis silking stages significantly reduced rows per ear and kernels per row, while it had insignificant effects on 100-kernel weight [43]. Our results show that rows per ear under CK and DI were significantly lower than under DI and DF in the 2017 and 2018 growing years, while rows per ear showed insignificant differences under CK, DI, and SF in 2019 and 2020 (Table 5). The reason for this is that maize suffered drought stress at the seeding and jointing stages in 2017 and at the jointing and tasseling stages in 2018 (Appendix A). DF ensured an on-demand supply of water and nitrogen during any growth period of summer maize, effectively reducing bald tip length and significantly increasing yield components compared with CK (Table 5). These results were consistent in that the optimization of water and nitrogen can significantly improve yield components, which in turn increase crop yields [12,29].

In summary, the increase in yield under drip-fertigated maize was mainly due to the fact that drip fertigation effectively avoided periodical drought and achieved an on-demand supply of nutrients, which increased the leaf area index, chlorophyll content, and photosynthetic rate. This further promoted aboveground biomass accumulation and translocation. Meanwhile, drip fertigation significantly prolonged the grain filling period duration, increased the 100-kernel grain weight, and ultimately led to a high grain yield.

## 5. Conclusions

In this 4-year field experiment, the increase in the leaf area index and photosynthetic rate of drip-fertigated maize promoted the accumulation of aboveground biomass. This provided a strong base for aboveground biomass translocation from pre-silking to grain yield. Drip fertigation significantly increased grain yield by 34.3% due to a higher accumulation and transportation of aboveground biomass compared with conventional irrigation and fertilization techniques. Additionally, drip fertigation significantly increased the grain filling rate of summer maize, especially in the middle and late stages of the grain filling processes, and prolonged the grain filling period, which jointly increased 100-kernel grain weight. Compared with conventional water and nitrogen management, 29.7%, 46.1%, and 24.2% grain yield increases were contributed by irrigation, nitrogen, and the coupling effects of irrigation and nitrogen, respectively. This study elucidates a higher grain yield mechanism under drip-fertigated maize in terms of an improved eco-physiology and phenology of the maize plant.

## Figures and Tables

**Figure 1 plants-13-01903-f001:**
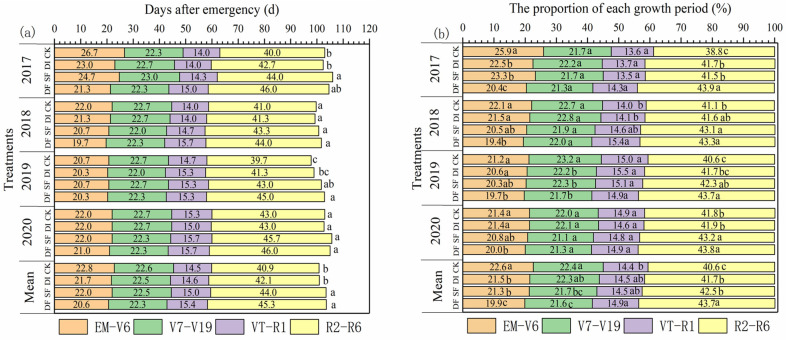
The response of irrigation and nitrogen management on maize phenology in the 2017–2020 growing seasons. Note: The left side (**a**) shows the time of each growth period, which is the number of days after emergence. The right side (**b**) shows the proportion of each reproductive period to the total reproductive period. Different lowercase letters represent significant differences between treatments.

**Figure 2 plants-13-01903-f002:**
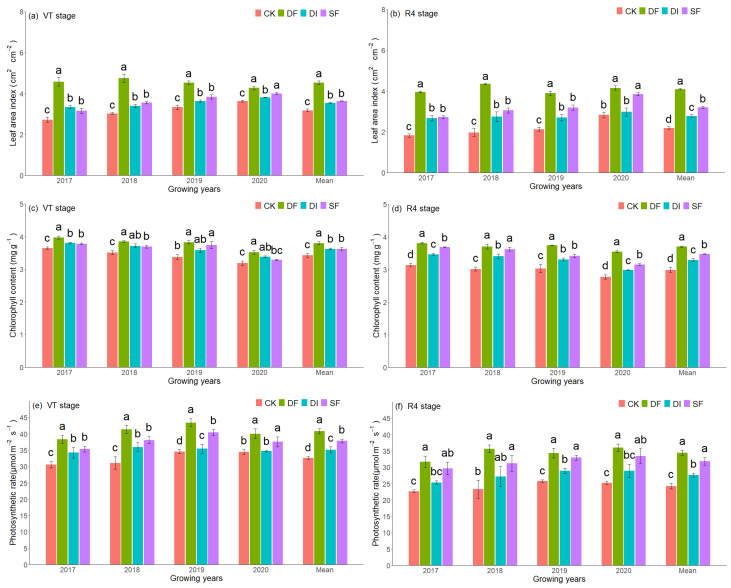
The response of leaf area index, chlorophyll content, and photosynthetic rate to irrigation and nitrogen management strategies. Different lowercase letters represent significant differences between treatments.

**Figure 3 plants-13-01903-f003:**
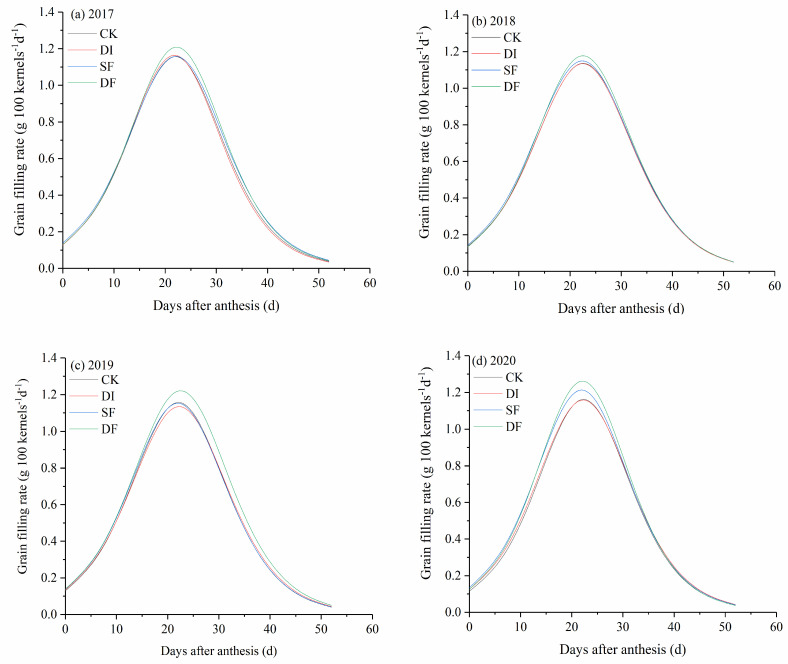
Dynamic change of grain filling rates under different irrigation and nitrogen strategies in the 2017–2020 growth seasons.

**Figure 4 plants-13-01903-f004:**
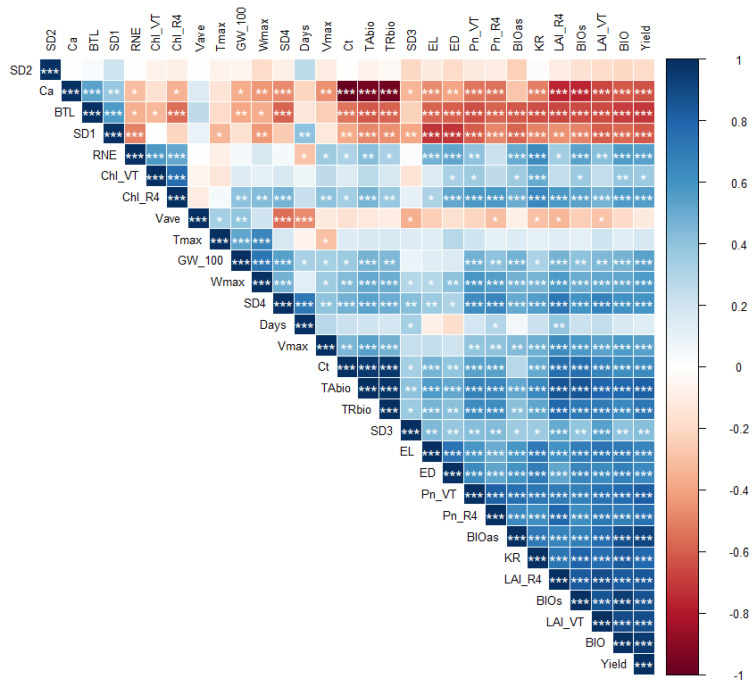
Correlation analysis among the traits under four treatments in four years. Note: SD1, SD2, SD3, SD4, and Days are the days of emergence-V6, V7-V19, VT-R1, R2-R6, and the entire growth period, respectively. LAI__VT_, Chl__VT_, and Pn__VT_ represent the leaf area index, chlorophyll content, and photosynthetic rate during the tasseling period, and LAI__R4_, Chl__R4_, and Pn__R4_ represent the leaf area index, chlorophyll content, and photosynthetic rate during the R4 stage. BTL, RNE, KR, GW_100, EL, and ED represent the bare top length, row numbers per ear, kernels per row, 100-kernel grain weight, ear length, and ear diameter. W_max_, T_max_, V_max_, and V_ave_ are the grain weight of the maximum grain filling rate, the occurrence time of the maximum grain filling rate, the maximum grain filling rate, and the average grain filling rate, respectively. BIO_s_, BIO_as_, and BIO represent the biomass before silking, after silking, and aboveground biomass. TA_bio_, TR_bio_, and Yield represent the biomass transport amount, rate, and grain yield of summer maize, respectively. * means a significant difference (*p* < 0.05); ** means a highly significant difference (*p* < 0.01); and *** means an extremely significant difference (*p* < 0.001).

**Table 1 plants-13-01903-t001:** Significant level of the treatments on leaf area index, chlorophyll content and photosynthetic rate of summer maize.

Factors	LAI__VT_	LAI__R4_	Chl__VT_	Chl__R4_	Pn__VT_	Pn__R4_
Irrigation (I)	**	**	**	**	**	**
Nitrogen (N)	**	**	**	**	**	**
Year (Y)	**	**	**	**	**	*
I × N	**	**	ns	ns	ns	ns
I × Y	**	**	ns	ns	ns	ns
N × Y	**	ns	ns	ns	ns	ns
I × N × Y	ns	ns	ns	*	ns	ns

Note: LAI__VT_, Chl__VT_, and Pn__VT_ represent the leaf area index, chlorophyll content, and photosynthetic rate during the tasseling period, and LAI__R4_, Chl__R4_, and Pn__R4_ represent the leaf area index, chlorophyll content, and photosynthetic rate during the R4 stage. * means a significant difference (*p* < 0.05); ** means a highly significant difference (*p* < 0.01); and ns means an insignificant difference (*p* > 0.05).

**Table 2 plants-13-01903-t002:** Grain filling parameters of summer maize under different irrigation and nitrogen fertilization management strategies in the 2017–2020 growing seasons.

Year	Treatments	W_max_	T_max_	V_max_	V_ave_
2017	CK	14.23 b	21.73 b	1.16 b	0.71 a
	DI	14.10 b	21.66 b	1.17 b	0.66 a
	SF	14.82 a	22.25 a	1.15 b	0.67 a
	DF	14.95 a	22.33 a	1.21 a	0.66 a
2018	CK	14.93 a	22.61 a	1.14 b	0.72 a
	DI	14.79 a	22.56 a	1.14 b	0.70 a
	SF	15.12 a	22.44 a	1.15 ab	0.69 a
	DF	15.22 a	22.63 a	1.18 a	0.69 a
2019	CK	14.58 b	21.14 a	1.16 b	0.70 a
	DI	14.61 b	22.26 a	1.14 b	0.68 ab
	SF	14.71 b	21.99 a	1.15 b	0.63 b
	DF	15.76 a	22.55 a	1.22 a	0.68 ab
2020	CK	14.19 b	22.43 a	1.14 b	0.65 a
	DI	14.81 ab	22.61 a	1.15 b	0.66 a
	SF	15.36 a	22.31 a	1.19 ab	0.65 a
	DF	15.45 a	22.19 a	1.25 a	0.65 a
Mean	CK	14.48 b	22.23 a	1.15 b	0.69 a
	DI	14.58 b	22.27 a	1.15 b	0.68 ab
	SF	15.00 a	22.25 a	1.16 b	0.66 b
	DF	15.34 a	22.42 a	1.21 a	0.67 ab

Note: W_max_ is the grain weight of the maximum grain filling rate; T_max_ is the occurrence time of the maximum grain filling rate; V_max_ is the maximum grain filling rate; and V_ave_ is the average grain filling rate. The values within a column and for the same year followed by different lowercase letters are significantly different at the *p* = 0.05 level.

**Table 3 plants-13-01903-t003:** Significance levels of the treatments on aboveground biomass, grain yield, and yield components of summer maize.

Factors	RNE	KR	GW__100_	EL	ED	BTL	BIO_m_	BIO_s_	BIO_as_	TA_bio_	TR_bio_	GY
Irrigation (I)	**	**	**	**	**	**	**	**	**	**	**	**
Nitrogen (N)	**	**	**	**	**	**	**	**	**	**	**	**
Year (Y)	**	**	ns	**	**	*	**	*	*	**	**	**
I × N	**	**	ns	*	ns	**	**	**	ns	**	ns	**
I × Y	**	**	ns	**	ns	**	**	**	ns	**	**	*
N × Y	ns	**	ns	ns	ns	*	ns	ns	ns	ns	ns	ns
I × N × Y	ns	ns	ns	ns	ns	**	ns	ns	ns	**	*	ns

Note: BTL, RNE, KR, GW__100_, EL, and ED represent the bare top length, rows number per ear, kernels per row, 100-kernels weight, ear length, and ear diameter. BIO_s_, BIO_as_, and BIO_m_ represent biomass at silking, after silking, and aboveground biomass. TA_bio_, TR_bio_, and GY represent the biomass transport amount, rate, and grain yield of summer maize, respectively. * means a significant difference (*p* < 0.05); ** means a highly significant difference (*p* < 0.01); and ns means an insignificant difference (*p* > 0.05).

**Table 4 plants-13-01903-t004:** Yield components of summer maize under different irrigation and nitrogen management strategies in the 2017–2020 growing seasons.

Year	Treatment	Rows per Ear	Kernels per Row	100-Kernel Weight (g)	Ear Length (cm)	Ear Diameter (cm)	Bare Top Length (cm)
2017	CK	14.6 c	27.3 d	28.0 c	13.6 b	4.2 b	2.3 a
	DI	15.5 b	31.7 b	28.3 bc	15.8 a	5.2 a	0.7 b
	SF	14.4 c	30.0 c	29.3 ab	14.3 b	5.1 a	1.0 b
	DF	16.3 a	35.4 a	30.2 a	16.7 a	5.6 a	0.7 b
2018	CK	15.3 b	29.1 c	28.3 c	15.1 b	5.0 b	1.4 a
	DI	16.1 a	31.2 b	29.1 bc	15.8 b	5.5 b	0.9 b
	SF	15.3 b	30.2 bc	29.7 ab	15.6 b	5.3 b	0.7 bc
	DF	16.4 a	33.9 a	30.2 a	16.8 a	6.1 a	0.6 c
2019	CK	15.5 b	29.9 c	27.7 b	16.2 b	5.5 b	1.8 a
	DI	15.4 b	29.9 c	28.9 ab	16.2 b	5.7 ab	0.9 b
	SF	15.9 b	34.1 b	27.5 b	16.2 b	5.7 ab	0.7 b
	DF	16.7 a	36.1 a	30.8 a	17.2 a	6.1 a	0.5 b
2020	CK	14.6 a	29.5 b	27.7 c	15.7 ab	5.0 b	1.1 a
	DI	14.3 a	29.8 b	29.7 ab	15.6 b	5.1 b	1.0 a
	SF	14.4 a	31.0 b	29.4 b	15.7 ab	4.9 b	1.0 a
	DF	14.8 a	33.5 a	30.7 a	16.4 a	5.7 a	0.7 a
Mean	CK	15.0 b	29.0 c	27.9 c	15.2 c	4.9 c	1.6 c
	DI	15.3 b	30.6 b	29.0 b	15.9 b	5.4 b	0.9 b
	SF	15.0 b	31.6 b	29.0 b	15.4 c	5.3 b	0.8 b
	DF	16.1 a	34.7 a	30.5 a	16.8 a	5.9 a	0.6 a

Note: Different lowercase letters represent significant differences at *p* = 0.05 level between treatments.

**Table 5 plants-13-01903-t005:** Effect of drip irrigation, split-N fertilization, and their impact on grain yield (GY), biomass at maturity (BIO_m_), and contribution to the GY/BIO_m_ increase of summer maize.

Year	Treatment	GY (t ha^−1^)	GY Increase Rate (%)	Contribution to GY Increase (%)			BIOm (t ha^−1^)	BIOm Increase Rate (%)	Contribution to BIOm Increase (%)		
				I	N	I × N			I	N	I × N
2017	CK	7.6 c	—	—	—	—	15.6 c	—	—	—	—
	DI	8.8 b	15.9	100	—	—	18.1 b	16.7	100	—	—
	SF	8.3 b	10.3	—	100	—	17.2 b	10.7	—	100	—
	DF	10.6 a	40.7	39.1	25.3	35.6	21.1 a	35.8	46.5	30.0	23.5
2018	CK	8.3 c	—	—	—	—	16.8 c	—	—	—	—
	DI	9.3 b	11.5	100	—	—	18.4 b	9.4	100	—	—
	SF	9.5 b	14.2	—	100	—	18.4 b	9.7	—	100	—
	DF	11.7 a	39.8	28.9	35.8	35.3	21.8 a	30.0	31.2	32.4	36.4
2019	CK	8.2 c	—	—	—	—	17.1 c	—	—	—	—
	DI	9.2 bc	11.8	100	—	—	18.1 c	5.5	100	—	—
	SF	9.8 b	19.7	—	100	—	19.6 b	14.3	—	100	—
	DF	11.3 a	37.4	31.4	52.7	15.8	22.3 a	29.8	18.4	48.0	33.6
2020	CK	8.7 c	—	—	—	—	17.6 b	—	—	—	—
	DI	9.0 bc	2.5	100	—	—	18.0 b	2.3	100	—	—
	SF	10.0 ab	14.8	—	100	—	18.9 ab	7.6	—	100	—
	DF	10.5 a	20.6	12.2	71.8	16.0	19.8 a	12.7	17.9	59.6	22.6
Mean	CK	8.2 c	—	—	—	—	16.8 c	—	—	—	—
	DI	9.0 b	10.2	100	—	—	18.1 b	8.2	100	—	—
	SF	9.5 b	15.8	—	100	—	18.5 b	10.6	—	100	—
	DF	11.0 a	34.3	29.7	46.1	24.2	21.3 a	26.8	30.7	39.5	29.9

Note: Different lowercase letters represent significant differences at *p* = 0.05 level between treatments.

**Table 6 plants-13-01903-t006:** Accumulation and transport of summer maize aboveground biomass and its contribution to grain yield under different irrigation and nitrogen management strategies in the 2017–2020 growing years.

Year	Treatment	BIO_s_ (t ha^−1^)	BIO_as_ (t ha^−1^)	TA_bio_ (t ha^−1^)	TR_bio_ (%)	C_t_ (%)	C_a_ (%)
2017	CK	8.8 c	6.7 c	0.8 b	9.2 b	10.8 b	89.2 a
	DI	10.5 b	7.6 b	1.1 b	10.8 b	12.9 b	87.1 a
	SF	9.8 bc	7.5 bc	0.9 b	9.0 b	10.6 b	89.4 a
	DF	12.5 a	8.7 a	2.0 a	15.9 a	18.7 a	81.3 b
2018	CK	9.5 c	7.2 b	1.1 c	11.4 c	13.1 b	86.9 a
	DI	10.8 b	7.6 b	1.7 b	16.0 b	18.6 ab	81.4 ab
	SF	10.6 b	7.8 b	1.7 b	16.0 b	17.8 ab	82.2 ab
	DF	12.9 a	9.0 a	2.7 a	21.0 a	23.2 a	76.8 b
2019	CK	9.9 c	7.3 b	0.9 b	9.5 b	11.4 b	88.6 a
	DI	10.0 c	8.0 ab	1.1 b	11.3 b	12.4 b	87.6 a
	SF	10.9 b	8.7 a	1.1 b	10.4 b	11.7 b	88.3 a
	DF	13.1 a	9.2 a	2.1 a	16.2 a	19.0 a	81.0 b
2020	CK	10.3 b	7.3 a	1.4 c	13.9 c	16.4 ab	83.6 ab
	DI	10.6 b	7.5 a	1.6 c	14.9 c	17.7 b	82.3 a
	SF	11.5 a	8.1 a	2.1 b	22.4 a	25.7 a	74.3 b
	DF	11.4 a	8.4 a	2.6 a	18.7 b	20.3 b	79.7 a
Mean	CK	9.6 c	7.1 c	1.1 c	11.0 c	12.9 c	87.1 a
	DI	10.5 b	7.7 bc	1.4 b	13.3 b	15.4 bc	84.6 ab
	SF	10.7 b	8.0 b	1.6 b	14.5 b	16.4 b	83.6 b
	DF	12.5 a	8.8 a	2.2 a	17.9 a	20.3 a	79.7 c

Note: BIO_s_ is the aboveground biomass at silking; BIO_as_ is the accumulated aboveground biomass after silking; TA_bio_ is the transport amount of aboveground biomass to grain yield; TR_bio_ is the transport ration of biomass to grain yield; C_t_ is the contribution of transported biomass to grain yield; and C_a_ is the contribution of accumulated biomass after silking to grain yield. Different lowercase letters represent significant differences at *p* = 0.05 level between treatments.

## Data Availability

Data are contained within the article and Appendix A.

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
