# Peer review of "Drip Fertigation Increases Maize Grain Yield by Affecting Phenology, Grain Filling Process, Biomass Accumulation and Translocation: A 4-Year Field Trial"

_plants, 2024, doi:10.3390/plants13141903_

Round 1

Reviewer 1 Report

Comments and Suggestions for Authors

This article examines how drip fertigation affects maize grain yield in terms of phenology, grain filling process biomass accumulation and translocation. The paper certainly meets the aim and the scope, as well as the high academic standards of the ‘Plants’ Journal. However, the following specific improvements should be made, before accepting the paper for possible publication to the Journal:

·        Material and methods well presented

·        Figure 1 pg 7/17 a short description must be placed

·        Figure 3, the first diagram must be corrected

·        Grain filling characteristed (section 3.3) you have to clearly state which had the best

·       The conclusions section could be further enriched to maximize the impact of your findings and maybe to address if there are limitations

Author Response

Reviewer #1

This article examines how drip fertigation affects maize grain yield in terms of phenology, grain filling process biomass accumulation and translocation. The paper certainly meets the aim and the scope, as well as the high academic standards of the ‘Plants’ Journal. However, the following specific improvements should be made, before accepting the paper for possible publication to the Journal:

Response: Thank you very much for your recognition. Your suggestions have played a significant role in improving the quality of the manuscript. We would like to thank you again for taking the time to review this manuscript.

  1. Material and methods well presented, Figure 1 pg 7/17 a short description must be placed

Response: We have added detailed description for this figure. Please see lines 218-220.

  1. Figure 3, the first diagram must be corrected

Response: We have corrected in this revised manuscript. Please see Figure 3.

  1. Grain filling characteristed (section 3.3) you have to clearly state which had the best

Response: We have added some description for grain filling processes. Please see lines 253-254.

  1. The conclusions section could be further enriched to maximize the impact of your findings and maybe to address if there are limitations

Response: We have revised the conclusion in the revised manuscript. Please see lines 430-441.

Reviewer 2 Report

Comments and Suggestions for Authors

This is a very good report of a useful study with clear results, especially reliable from the number of seasons studied. 

My only suggestions are to expand on the methods description:  what is the plot size, how many replicate plots, and what is the planting density?  Fifteen ears per plot for yield determination seems low, but that would depend on the number of plots, which was not given.  Does this variety produce only one ear per plant?

If the researcher does not know in advance how often irrigation will be needed, how does one decide how much fertilizer to apply at each drip irrigation?  It was not clear to me how this was managed.

Comments on the Quality of English Language

This paper is easily understandable, but it could be more polished. 

Author Response

Reviewer #2

  1. This is a very good report of a useful study with clear results, especially reliable from the number of seasons studied. 

Response: Thank you very much for your recognition. Your suggestions have played a significant role in improving the quality of the manuscript. We would like to thank you again for taking the time to review this manuscript.

  1. My only suggestions are to expand on the methods description:  what is the plot size, how many replicate plots, and what is the planting density?  Fifteen ears per plot for yield determination seems low, but that would depend on the number of plots, which was not given.  Does this variety produce only one ear per plant?

Response: According to your suggestion, we have added related information in the revised manuscript. Please see lines 116-132.

  1. If the researcher does not know in advance how often irrigation will be needed, how does one decide how much fertilizer to apply at each drip irrigation?  It was not clear to me how this was managed.

Response: This is a good question. In this experiment, we developed a fertilization regime, i.e. 20%, 30%, 30% and 20% nitrogen were fertilized at sowing, seedling, jointing and grain filling stage, respectively. If irrigation is not required within one week of reaching the fertilization growth period, we will specifically fertilize, which is why some fertilization and irrigation times are inconsistent (Table S2). The amount of irrigation used for each fertilization is 10mm.